# Brief communication: A note on the variance of wind speed and turbulence intensity

Cristina L. Archer[1, 2]

[1]Center for Research in Wind (CReW), University of Delaware, Delaware, USA
[2]Department of Environmental, Land, and Infrastructure Engineering (DIATI), Politecnico di Torino, Turin, Italy

**Correspondence:** Cristina L. Archer (carcher@udel.edu)

**Abstract.** This note addresses the issue that several papers in the peer-reviewed literature of wind energy applications have used an incorrect equation that equals the variance of wind speed ($\sigma_U^2$) to the sum of the variances of the wind components. This incorrect equation is often used to calculate turbulence intensity (TI), which, as a consequence, is often incorrectly estimated too. While exact analytical equations do not exist, here two approximate analytical equations are derived for $\sigma_U^2$ and TI as functions of the variances and means of the wind components. Both formulations are validated with samples from a prior field campaign and perform satisfactorily.

## 1 Introduction

The standard deviation of wind speed, which is the square root of the variance, is an important parameter in meteorology and in wind energy applications because it is a measure of wind variability. In the International Electrotechnical Commission (IEC) standard (International Electrotechnical Commission, 2019) that wind turbines must comply with, the standard deviation of wind speed is part of the definition of turbulence intensity (TI), which is "the ratio of the wind speed standard deviation to the mean wind speed, determined from the same set of measured data samples of wind speed, and taken over a specified period of time". The issue of how to calculate the variance of the three-dimensional (3D) wind vector is, however, not straightforward if the high-frequency raw data are not available.

The first problem is the system of coordinates. Since wind turbines always face the wind, especially in the first experiments that were conducted in wind tunnels and in idealized simulations, the convention in wind engineering has always been to align the $x$-axis along the mean wind direction. This convention of rotating the axes so that the $x$-axis would align with the mean wind direction is also adopted in boundary-layer meteorology, micrometeorology, and air pollution science, due to the focus on turbulence (Kaimal and Finnigan, 1994). The rotated system is also adopted in the IEC standard, which defines the three components of the turbulent wind velocity vector as longitudinal (along the direction of the mean wind velocity), lateral (horizontal and normal to the longitudinal direction), and upward (normal to both the longitudinal and lateral directions), with "turbulence standard deviations" called $\sigma_1, \sigma_2$, and $\sigma_3$, respectively. With this convention, the variance of wind speed is accurately approximated as (but not exactly equal to) the variance of the $u$-component of the wind, i.e., $\sigma_u^2$ (or $\sigma_1^2$ in the IEC standard).

By contrast, in mesoscale meteorology and, more broadly, in geophysical applications, such as meteorological field campaigns or simulations of weather events, the convention is to align the $x$-axis along the east-west direction (and the $y$-axis along the north-south). The third system of coordinates is a simple Cartesian one, with fixed and orthogonal $x, y$, and $z$ axes, none of which necessarily aligned with the mean wind direction. It is often used in idealized numerical simulations for weather and climate applications, and, at times, in large-eddy simulations of flows past wind turbines.

With the Cartesian and the geophysical systems of coordinates, the variance of wind speed $\sigma_U^2$ is no longer accurately approximated as the variance of the $u$-component. Furthermore, an exact equation for the wind speed variance as a function of the variances (and the means) of the wind components alone is impossible to obtain analytically for the fixed coordinate systems, because of the non-linear function (i.e., square root of the sum of the squares) that relates the magnitude of the wind vector $U$ to its components $u, v$, and $w$ along the $x, y$, and $z$ axis, respectively (discussed later in Eq. 1 and 6). In summary, the relationship between the variance of wind speed and those of the wind components depends on the system of coordinates and therefore confusion can arise among disciplines because of their different axis conventions.

The second problem is that of internal and external inconsistencies in the IEC standard. While the IEC standard clearly defines TI as the "the ratio of the wind speed standard deviation to the mean wind speed" in the "Terms and definitions" section, in later sections it actually appears to use $\sigma_1$, not $\sigma_U$, to define normal turbulence conditions and for fatigue load calculations (e.g., their Eq. 11). This would imply, wrongfully, that only the longitudinal fluctuations of the wind vector are relevant to a wind turbine performance. Also, the IEC standard is possibly the only case in which a single value of turbulence intensity is adopted. In most fields, including wind systems engineering, three turbulence intensities are typically used, one for each direction ($TI_x = \sigma_u / \bar{U}$, and similarly for $TI_y$ and $TI_z$). Lastly, the IEC standard assumes explicitly that the "turbulence standard deviation $\sigma_1$ [...] shall be assumed to be invariant with height", while it is well known that there is a vertical gradient of TI in the atmospheric boundary layer, thus the turbulence fluctuations measured, for example, near the ground are not representative of those at hub height.

The third problem is the temporal scales that should be considered in the calculation of TI and $\sigma_U^2$. Strictly speaking, TI should refer only to fluctuations of the wind in the micro-scale (i.e., time averages of the order of minutes), thus to the right of the spectral gap in the wind spectrum. The IEC standard is clear in this respect: turbulence is defined as "random variations in the wind velocity from 10 min averages". By contrast, wind fluctuations associated with meso or synoptic scale features belong to the left of the spectral gap and should not be called turbulent. In such cases, the ratio of the wind speed standard deviation over the mean, calculated over longer time intervals (i.e., hours to days), can still be obtained, but it should not be called a "turbulence" intensity. Therefore, using these meso or synoptic scale fluctuations in the calculation of TI for wind energy applications, especially to comply with the IEC standard, should be done with extreme caution, or avoided altogether.

To further complicate the matter, an incorrect expression for the variance of wind speed is often found in the literature, namely, the sum of the variances of the wind components, and often treated, incorrectly, as an exact definition [see for example Eq. 6 in Joffre and Laurila (1988)]. There is no theoretical or statistical justification for this incorrect expression and no special case (e.g., independent or uncorrelated variables, or a specific statistical distribution, or particular spatial conditions) for which it would apply.

This note addresses the incorrect formulation issue by proposing an analytical approximation for the wind speed variance and one for turbulence intensity (as defined in the IEC standard), to be used with any system of coordinates in cases when only the variances (and the means) of the wind components are available from measurements or simulations. The equations derived here may be applied to any temporal scale, but the focus is on the micro-scale.

## 2 Definitions

The equations derived hereafter are valid for any coordinate system (e.g., simple Cartesian, rotated, or geophysical). For the sake of generality, let us start with the simple Cartesian system, for which the three axes are fixed (i.e, not rotated to align $x$ with the mean wind direction or with the west-east direction). The wind components along $x, y$ and $z$ are $u, v$ and $w$, respectively, and the magnitude $U$ is a non-linear function of all three:

$$U = f(u, v, w) = \sqrt{u^2 + v^2 + w^2}. \tag{1}$$

The means $\bar{u}$, $\bar{v}$, $\bar{w}$, and $\bar{U}$, calculated over a set of $N$ measurements $u_t$, $v_t$, $w_t$, and $U_t$, each taken at time $t$, are:

$$\bar{u} = \frac{1}{N} \sum_t u_t, \quad \bar{v} = \frac{1}{N} \sum_t v_t, \quad \bar{w} = \frac{1}{N} \sum_t w_t, \tag{2}$$

$$\bar{U} = \frac{1}{N} \sum_t U_t. \tag{3}$$

The variances $\sigma_u^2$, $\sigma_v^2$, $\sigma_w^2$, and $\sigma_U^2$ are:

$$\sigma_u^2 = \frac{1}{N} \sum_t (u_t - \bar{u})^2 = \overline{(u - \bar{u})^2} = \overline{u^2 - 2u\bar{u} + \bar{u}^2} = \overline{u^2} - 2\bar{u}^2 + \bar{u}^2 = \overline{u^2} - \bar{u}^2, \tag{4}$$

$$\sigma_v^2 = \frac{1}{N} \sum_t (v_t - \bar{v})^2 = \overline{v^2} - \bar{v}^2, \quad \sigma_w^2 = \frac{1}{N} \sum_t (w_t - \bar{w})^2 = \overline{w^2} - \bar{w}^2, \tag{5}$$

$$\sigma_U^2 = \frac{1}{N} \sum_t \left(U_t - \bar{U}\right)^2 = \overline{U^2} - \bar{U}^2 = \overline{u^2} + \overline{v^2} + \overline{w^2} - \bar{U}^2 = \sigma_u^2 + \sigma_v^2 + \sigma_w^2 + \bar{u}^2 + \bar{v}^2 + \bar{w}^2 - \bar{U}^2. \tag{6}$$

Eq. 6 may not be simplified analytically any further because:

$$\bar{U}^2 = \left(\overline{\sqrt{u^2 + v^2 + w^2}}\right)^2 \neq \bar{u}^2 + \bar{v}^2 + \bar{w}^2. \tag{7}$$

As a consequence:

$$\sigma_U^2 \neq \sigma_u^2 + \sigma_v^2 + \sigma_w^2 \tag{8}$$

and

$$TI^2 = \frac{\sigma_U^2}{\bar{U}^2} \neq \frac{\sigma_u^2 + \sigma_v^2 + \sigma_w^2}{\bar{U}^2} = \frac{2\,TKE}{\bar{U}^2}, \tag{9}$$

where:

$$TKE = \frac{\sigma_u^2 + \sigma_v^2 + \sigma_w^2}{2}. \tag{10}$$

In order to obtain an expression for the variance of wind speed, we first need to recognize that the wind is intrinsically turbulent and therefore we can use the Reynolds averaging approach. The turbulent fluctuations, usually denoted with a prime ($'$), in this case coincide exactly with the differences from the means ($\delta$) as follows:

$$u_t = \bar{u} + u_t' = \bar{u} + \delta u_t, \tag{11}$$

and similarly for $v_t$, $w_t$, and $U_t$. Therefore the variances can be rewritten exactly as:

$$\sigma_u^2 = \frac{1}{N}\sum u_t'^2 = \frac{1}{N}\sum (\delta u_t)^2 = \overline{u'^2} = \overline{(\delta u)^2}, \tag{12}$$

and similarly for $\sigma_v^2$, $\sigma_w^2$, and $\sigma_U^2$.

## 3 Proposed formulation

Following the approach of Ackermann (1983) and Baird (1962), we introduce the only approximation of this manuscript: that the $\delta$'s coincide with the differentials. This is equivalent to assuming that the fluctuations (and the $\delta$'s) are smaller in magnitude than their respective means, which is realistic, but may or may not be true in all atmospheric conditions. The goal is to derive formulations for $\sigma_U^2$ and $TI$ that depend only on statistics of the wind components.

First, we use the assumption that the $\delta$'s can be approximated as differentials as follows:

$$(\delta U)^2 \approx \left(\frac{\partial U}{\partial u}\right)^2 (\delta u)^2 + \left(\frac{\partial U}{\partial v}\right)^2 (\delta v)^2 + \left(\frac{\partial U}{\partial w}\right)^2 (\delta w)^2 \tag{13}$$

$$+2\left(\frac{\partial U}{\partial u}\right)\left(\frac{\partial U}{\partial v}\right)\delta u \delta v + 2\left(\frac{\partial U}{\partial u}\right)\left(\frac{\partial U}{\partial w}\right)\delta u \delta w + 2\left(\frac{\partial U}{\partial v}\right)\left(\frac{\partial U}{\partial w}\right)\delta v \delta w.$$

Note that, in Eq. 13, the partial derivatives are to be evaluated at the "point" of the function $U = f(u, v, w)$ around which there are the fluctuations, thus for the mean values $\bar{u}$, $\bar{v}$, and $\bar{w}$. The three partial derivatives are therefore:

$$\left(\frac{\partial U}{\partial u}\right) = \frac{\partial U}{\partial u}\bigg|_{\bar{u},\bar{v},\bar{w}} = \frac{1}{2}\left(u^2 + v^2 + w^2\right)^{-\frac{1}{2}} (2u)\bigg|_{\bar{u},\bar{v},\bar{w}} = \frac{\bar{u}}{\sqrt{\bar{u}^2 + \bar{v}^2 + \bar{w}^2}}, \tag{14}$$

$$\left(\frac{\partial U}{\partial v}\right) = \frac{\partial U}{\partial v}\bigg|_{\bar{u},\bar{v},\bar{w}} = \frac{1}{2}\left(u^2 + v^2 + w^2\right)^{-\frac{1}{2}} (2v)\bigg|_{\bar{u},\bar{v},\bar{w}} = \frac{\bar{v}}{\sqrt{\bar{u}^2 + \bar{v}^2 + \bar{w}^2}}, \tag{15}$$

$$\left(\frac{\partial U}{\partial w}\right) = \frac{\partial U}{\partial w}\bigg|_{\bar{u},\bar{v},\bar{w}} = \frac{1}{2}\left(u^2 + v^2 + w^2\right)^{-\frac{1}{2}}(2w)\bigg|_{\bar{u},\bar{v},\bar{w}} = \frac{\bar{w}}{\sqrt{\bar{u}^2 + \bar{v}^2 + \bar{w}^2}}, \tag{16}$$

which are not a function of time $t$. Replacing Eqs. 14–16 into Eq. 13 leads to the following expression for $\sigma_U^2$:

$$\sigma_U^2 = \frac{1}{N}\sum(\delta U)^2 = \overline{(\delta U)^2} \approx \frac{\bar{u}^2\sigma_u^2 + \bar{v}^2\sigma_v^2 + \bar{w}^2\sigma_w^2 + 2\bar{u}\bar{v}\sigma_{uv} + 2\bar{u}\bar{w}\sigma_{uw} + 2\bar{v}\bar{w}\sigma_{vw}}{\bar{u}^2 + \bar{v}^2 + \bar{w}^2}, \tag{17}$$

where $\sigma_{uv}, \sigma_{uw}$, and $\sigma_{vw}$ are the covariances of $u$ and $v$, $u$ and $w$, and $v$ and $w$, respectively, which can be positive or negative.

To obtain an expression for $TI$, we derive an approximation for $\overline{U}$ as follows:

$$\bar{U} = \overline{\sqrt{(\bar{u}+u')^2 + (\bar{v}+v')^2 + (\bar{w}+w')^2}} \tag{18}$$

$$= \sqrt{\bar{u}^2 + \bar{v}^2 + \bar{w}^2}\overline{\sqrt{\frac{(\bar{u}+u')^2 + (\bar{v}+v')^2 + (\bar{w}+w')^2}{\bar{u}^2 + \bar{v}^2 + \bar{w}^2}}} \tag{19}$$

$$= \sqrt{\bar{u}^2 + \bar{v}^2 + \bar{w}^2}\overline{\sqrt{1 + \frac{2u'\bar{u} + u'^2 + 2v'\bar{v} + v'^2 + 2w'\bar{w} + w'^2}{\bar{u}^2 + \bar{v}^2 + \bar{w}^2}}}. \tag{20}$$

The term under the square root can be simplified via the binomial approximation for $\alpha = 1/2$:

$$(1+x)^\alpha \approx (1+\alpha x), \tag{21}$$

valid for $|x| < 1$ and $|\alpha x| << 1$, which are generally true in Eq. 20 due to the assumption that the fluctuations are small with respect to the means, as follows:

$$\bar{U} \approx \sqrt{\bar{u}^2 + \bar{v}^2 + \bar{w}^2}\,\overline{1 + \frac{u'\bar{u} + v'\bar{v} + w'\bar{w}}{\bar{u}^2 + \bar{v}^2 + \bar{w}^2} + \frac{1}{2}\frac{u'^2 + v'^2 + w'^2}{\bar{u}^2 + \bar{v}^2 + \bar{w}^2}}. \tag{22}$$

Using the Reynolds averaging properties, the final expressions for $\overline{U}$ and $\overline{U}^2$ are:

$$\bar{U} \approx \sqrt{\bar{u}^2 + \bar{v}^2 + \bar{w}^2}\left(1 + \frac{1}{2}\frac{\sigma_u^2 + \sigma_v^2 + \sigma_w^2}{\bar{u}^2 + \bar{v}^2 + \bar{w}^2}\right), \tag{23}$$

$$\bar{U}^2 \approx \left(\bar{u}^2 + \bar{v}^2 + \bar{w}^2\right)\left(1 + \frac{1}{2}\frac{\sigma_u^2 + \sigma_v^2 + \sigma_w^2}{\bar{u}^2 + \bar{v}^2 + \bar{w}^2}\right)^2. \tag{24}$$

Since the term in parenthesis in Eq. 24 is greater than 1, not only is the inequality in Eq. 7 confirmed, but it can be further expanded to:

$$\bar{U}^2 > \bar{u}^2 + \bar{v}^2 + \bar{w}^2. \tag{25}$$

One could be tempted to replace the expression for $\bar{U}^2$ from Eq. 24 in Eq. 6, but doing so would cause the expression for the variance of wind speed to become negative because the error introduced by the binomial approximation, although small when

used for $\bar{U}$, is amplified in $\bar{U}^2$, especially when it is used in a difference of terms of similar magnitudes as in Eq. 6. When used in the denominator and alone, however, as is the case for $TI$ from Eq. 9, Eq. 24 is acceptable and we obtain:

$$TI^2 = \frac{\sigma_U^2}{\overline{U}^2} \approx \frac{\bar{u}^2\sigma_u^2 + \bar{v}^2\sigma_v^2 + \bar{w}^2\sigma_w^2 + 2\bar{u}\bar{v}\sigma_{uv} + 2\bar{u}\bar{w}\sigma_{uw} + 2\bar{v}\bar{w}\sigma_{vw}}{(\bar{u}^2+\bar{v}^2+\bar{w}^2)^2 \left(1 + \frac{1}{2}\frac{\sigma_u^2+\sigma_v^2+\sigma_w^2}{\bar{u}^2+\bar{v}^2+\bar{w}^2}\right)^2}. \tag{26}$$

To simplify the notation without loosing generality, we hereafter assume that the wind is a two-dimensional (2D) vector. This assumption is often used in mesoscale meteorology and is needed when only 2D measurements of the wind are available (e.g., with a cup anemometer). Thus all terms that are a function of $w$ drop from Eq. 17:

$$\sigma_U^2 \approx \frac{\bar{u}^2\sigma_u^2 + \bar{v}^2\sigma_v^2 + 2\bar{u}\bar{v}\sigma_{uv}}{\bar{u}^2+\bar{v}^2} < \sigma_u^2 + \sigma_v^2. \tag{27}$$

Using $\sigma_u^2 + \sigma_v^2$ as an approximation for $\sigma_U^2$ generally causes an over-estimation of the variance of $U$, especially when $\bar{u}$ and $\bar{v}$

are of opposite sign (e.g., in the second and fourth quadrants) and the covariance is positive, or vice versa when $\bar{u}$ and $\bar{v}$ are of the same sign and $\sigma_{uv}$ is negative.

If the two variables $u, v$ were independent (but they are not), their covariance $\sigma_{uv}$ would be zero; since $\sigma_{uv}$ is often unknown, it can be set to zero as an approximation, to give an expression that is still overestimated by the sum of the wind component variances:

$$\sigma_U^2 \approx \frac{\bar{u}^2\sigma_u^2 + \bar{v}^2\sigma_v^2}{\bar{u}^2+\bar{v}^2} < \sigma_u^2 + \sigma_v^2. \tag{28}$$

Similarly for $TI$ with 2D wind vectors, Eq. 26 becomes:

$$TI^2 \approx \frac{\bar{u}^2\sigma_u^2 + \bar{v}^2\sigma_v^2 + 2\bar{u}\bar{v}\sigma_{uv}}{(\bar{u}^2+\bar{v}^2)^2 \left(1 + \frac{1}{2}\frac{\sigma_u^2+\sigma_v^2}{\bar{u}^2+\bar{v}^2}\right)^2} < \frac{\sigma_u^2 + \sigma_v^2}{\bar{u}^2+\bar{v}^2}. \tag{29}$$

If the approximation for $\sigma_U^2$ from Eq. 28 and that for $\bar{U}$ from Eq. 7 are used, then:

$$TI^2 \approx \frac{\bar{u}^2\sigma_u^2 + \bar{v}^2\sigma_v^2}{(\bar{u}^2+\bar{v}^2)^2} < \frac{\sigma_u^2 + \sigma_v^2}{\bar{u}^2+\bar{v}^2}. \tag{30}$$

Note that, when the $x$-axis is rotated in such a way that it is aligned along the mean wind, $\bar{v} \approx 0$ and therefore $\sigma_U^2 \approx \sigma_u^2$ from Eq. 27, consistent with the IEC convention (in which it is called $\sigma_1^2$) and further supported in the derivation in Appendix B by Larsén (2022). In this rotated coordinate system with the $x$-axis aligned with the mean wind, a better alternative to Eq. 23 for $\bar{U}$ in 2D is the approximation from Kristensen (1998):

$$\bar{U} = \bar{u} + \frac{\sigma_v^2}{2\bar{u}^2}, \tag{31}$$

thus $TI$ can be approximated as:

$$TI^2 = \frac{\sigma_u^2}{(\bar{u} + \frac{\sigma_v^2}{2\bar{u}^2})^2}. \tag{32}$$

## 4 Application

Wind measurements collected with a 20-Hz sonic anemometer mounted at 4 m during the American WAKE experimeNt (AWAKEN) field campaign (Atmosphere to Electrons (A2e), 2025), conducted in northern Oklahoma (U.S.A.) around five wind farms between 2022 and 2024, are used to demonstrate the validity of the proposed formulations and compare their performance against that of the inexact equations discussed above. A one-week period (23–29 July 2023) is selected for the analysis (Figure 1d).

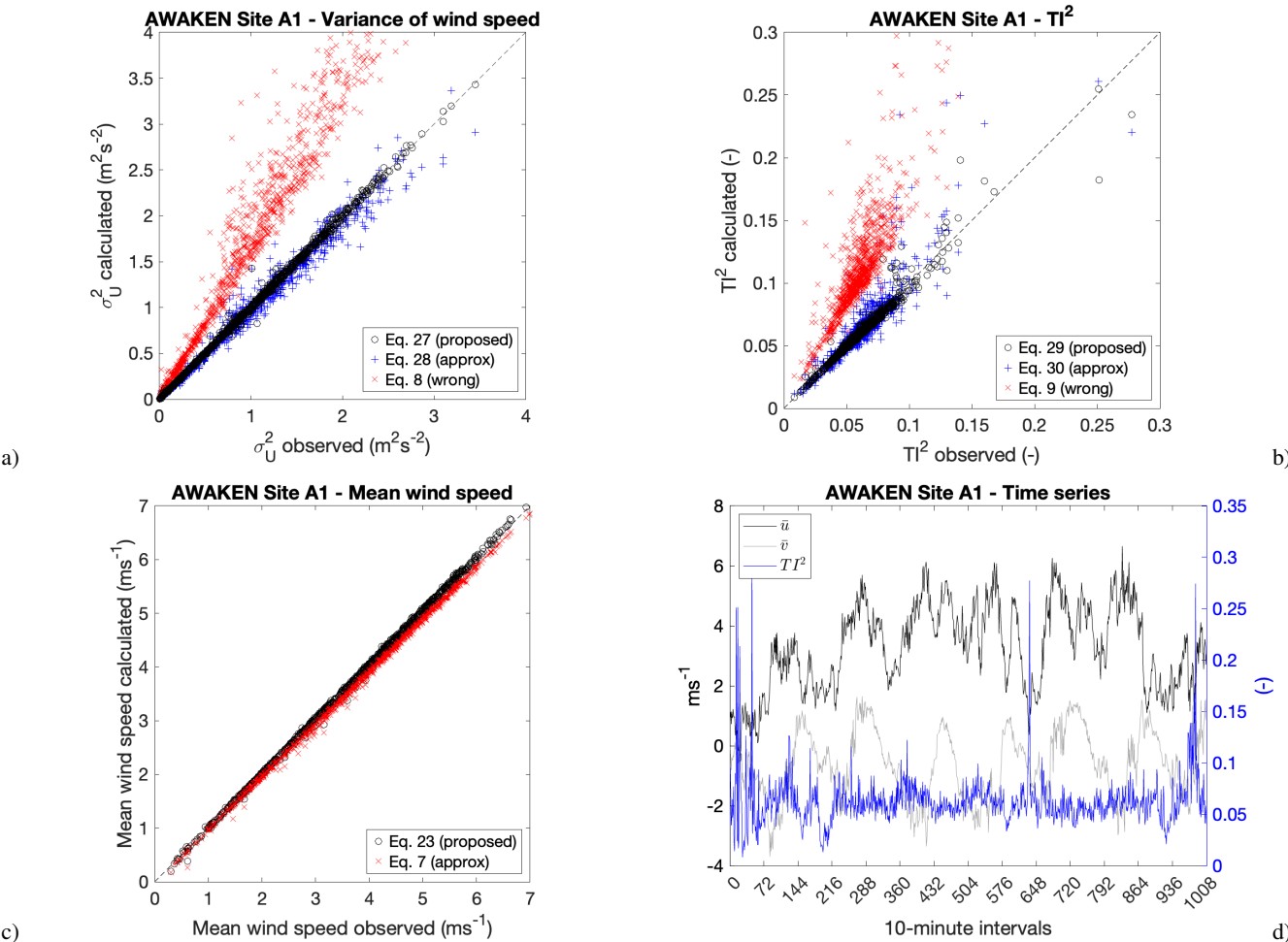

**Figure 1.** Scatter plots of 10-minute statistics from the AWAKEN campaign during the week of 23–29 July 2023: a) wind speed variance; b) turbulence intensity (squared); and c) mean wind speed. The time series of observed mean wind components and turbulence intensity (squared) are in d).

The proposed formulations for $\sigma_U^2$ (Eq. 27), $TI^2$ (Eq. 29), and $\bar{U}$ (Eq. 23) perform very well, with a very close alignment with the 1:1 line (Figure 1, a–c). For the variance, the mean absolute percent error (MAPE) is 2.4% for the proposed formulation,

while using $\sigma_u^2 + \sigma_v^2$ (Eq. 28) always causes an overestimation (i.e., positive error), with a MAPE of 78.6% and a large positive bias of 0.70 m$^2$s$^{-2}$ (Table 1). The MAPE for $TI^2$ with Eq. 29 is 3.7%, slightly larger than that for $\sigma_U^2$, due to the additional approximation introduced by the division of Eq. 27 by Eq. 24. $TI$ is always grossly overestimated by using the approximation from Eq. 9 (MAPE = 95.1%), because the numerator overestimates, while the denominator slightly underestimates.

**Table 1.** Error analysis of the various equations analyzed in the manuscript for the AWAKEN campaign during the week of 23–29 July 2023.

| EQUATION | | BIAS | RMSE | MAPE |
|---|---|---|---|---|
| Eq. 27 (proposed): | $\sigma_U^2 \approx \dfrac{\bar{u}^2\sigma_u^2 + \bar{v}^2\sigma_v^2 + 2\bar{u}\bar{v}\sigma_{uv}}{\bar{u}^2 + \bar{v}^2}$ | $1.2\times10^{-3}$ m$^2$s$^{-2}$ | 0.03 m$^2$s$^{-2}$ | 2.4% |
| Eq. 28 (approx): | $\sigma_U^2 \approx \dfrac{\bar{u}^2\sigma_u^2 + \bar{v}^2\sigma_v^2}{\bar{u}^2 + \bar{v}^2}$ | -0.03 m$^2$s$^{-2}$ | 0.10 m$^2$s$^{-2}$ | 7.9% |
| Eq. 8 (wrong): | $\sigma_U^2 \neq \sigma_u^2 + \sigma_v^2$ | 0.70 m$^2$s$^{-2}$ | 0.88 m$^2$s$^{-2}$ | 78.6% |
| Eq. 29 (proposed): | $TI^2 \approx \dfrac{\bar{u}^2\sigma_u^2 + \bar{v}^2\sigma_v^2 + 2\bar{u}\bar{v}\sigma_{uv}}{(\bar{u}^2 + \bar{v}^2)^2 \left(1 + \dfrac{1}{2}\dfrac{\sigma_u^2 + \sigma_v^2}{\bar{u}^2 + \bar{v}^2}\right)^2}$ | $-3.3\times10^{-4}$ | 0.01 | 3.7% |
| Eq. 30 (approx): | $TI^2 \approx \dfrac{\bar{u}^2\sigma_u^2 + \bar{v}^2\sigma_v^2}{(\bar{u}^2 + \bar{v}^2)^2}$ | $4.1\times10^{-3}$ | 0.04 | 10.5% |
| Eq. 9 (wrong): | $TI^2 \neq \dfrac{\sigma_u^2 + \sigma_v^2}{(\bar{u}^2 + \bar{v}^2)^2}$ | 0.06 | 0.18 | 95.1% |
| Eq. 23 (proposed): | $\bar{U} \approx \sqrt{\bar{u}^2 + \bar{v}^2}\left(1 + \dfrac{1}{2}\dfrac{\sigma_u^2 + \sigma_v^2}{\bar{u}^2 + \bar{v}^2}\right)$ | 0.03 ms$^{-1}$ | 0.05 ms$^{-1}$ | 1.2% |
| Eq. 7 (approx): | $\bar{U} \approx \sqrt{\bar{u}^2 + \bar{v}^2}$ | -0.09 ms$^{-1}$ | 0.10 ms$^{-1}$ | 2.5% |

## 5  Conclusions

An analytical equation that approximates the variance of wind speed as a function of the variances and the means of the wind components is derived for any coordinate system (e.g., Cartesian, rotated, or geophysical), under the only assumption that the turbulent fluctuations of the wind components are small with respect to their means. The approximation for the variance of wind speed is then used, after a few steps, to derive another approximation for turbulence intensity. Although a thorough validation is beyond the scope of this note, both formulations appear to perform well for a few samples of observations obtained 175 during the AWAKEN field campaign of 2023 and to outperform the two incorrect equations that have been used at times in the literature.

*Competing interests.*  Archer is a member of the editorial board of Wind Energy Science.

*Acknowledgements.* The author would like to thank Dr. Costantino Manes of the Department of Environment, Land and Infrastructure Engineering (DIATI) of the Politecnico of Torino (Italy) and Dr. Jakob Mann of the Department of Wind and Energy Systems of the Danish Technical University (DTU, Denmark) for the excellent discussions related to the content of this manuscript. Productive exchanges with the Associate Editor of this article, Dr. Etienne Cheynet, are also acknowledged.

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
