# Peer review of "Brief communication: A note on the variance of wind speed and turbulence intensity"

_Wind Energy Science, 2024_

## Author Comment (AC1)

**Reply to the Reviewer #1 of "Brief communication: A note on the variance of wind speed and turbulence intensity"**

December 16, 2024

Please note that the reviewers' comments are in *italic*, my responses in regular font, and the changes to the manuscript in blue color.

- *This paper deals with the difference between the variance of the wind component along the mean wind vector and the variance of the length of the wind vector, also called the wind speed. It is well known that those quantities are under most circumstances (i.e. not too high turbulence intensity) almost equal (e.g. L.. Kristensen 1998, JTech, vol 5, p6). The transverse component enters only the speed variance to second order in the turbulence intensity (see eq 8 in the mentioned paper). These observations do not change if the coordinate system is not aligned with the wind.*

First of all, even if the variance of the wind speed and the variance of the mean wind vector were almost equal, what I am trying to convey in this note is that the two are not identical and, more importantly, that one should not be used as the definition for the other.

Second, the paper by Kristensen (1998) deals with the calibration of cup anemometers in wind tunnels under steady-state conditions. Eq. 8 in particular is:

$$U = \sqrt{u^2 + v^2} \approx \bar{u} + u' + \frac{v'^2}{2\bar{u}}. \tag{1}$$

The Reviewer cites this equation to support that transverse perturbations only affect the variance of wind speed to the second order; however, this is **not** an equation for the variance of wind speed, thus the point is not proven with this equation. Also, while in wind tunnels (a rather artificial setup) the transverse fluctuations might be smaller than those along the mean wind, atmospheric turbulence is generally considered to be homogeneous and isotropic, thus the statistics of turbulence, such as variances, must be the same along any direction and do not vary if the coordinate system is changed. We know that atmospheric turbulence is not isotropic along the vertical due to the presence of the ground, but turbulence isotropy is a well accepted hypothesis for the horizontal directions. As such, it is not possible that the transverse fluctuations be always smaller than those along the mean wind; they might be under certain circumstances (e.g., wind tunnels), but not always.

- *The other subject paper is an apparent mistake in the literature. The author states that the variance of the wind speed is sometimes mistakingly said to be equal to the sum of the variances of the two horizontal components. This is obviously wrong, as the author clearly states, but I'm am unaware of these mistakes in the literature. The author does not provide evidence for these mistakes, which makes the need for this paper limited. The author might be wary to point out mistakes in specific papers, but this is unfortunately what has to be done in order to advance science. You cannot leave it to the readers to find documentation for this possible mistake in the literature.*

I am indeed uncomfortable publishing a note that directly points out mistakes by fellow scientists. The point of my note is to provide a clear reference as to why the two variances are not the same. As such, I provide below a list of four papers with the above-mentioned error. My intention here is to satisfy the Reviewer's request for evidence, but I do not intend to add this list in the main document. Since

the entire review process is public in WES, it will be possible in the future to find this information anyway, but, as far as I am concerned, not in the main manuscript.

- Eq. 6 in Joffre and Laurila (1988);
- Eq. 1 in Mortarini et al. (2016);
- Eq. 1 in Bodini et al. (2020); and
- Eq. 11 in Klemmer et al. (2024).

**References**

Bodini, N., Lundquist, J. K., and Kirincich, A.: Offshore wind turbines will encounter very low atmospheric turbulence, Journal of Physics: Conference Series, 1452, 012 023, https://doi.org/10.1088/1742-6596/1452/1/012023, 2020.

Joffre, S. M. and Laurila, T.: Standard deviations of wind speed and direction from observations over a smooth surface, Journal of Applied Meteorology and Climatology, 27, 550 – 561, https://doi.org/10.1175/1520-0450(1988)027⟨0550:SDOWSA⟩2.0.CO;2, 1988.

Klemmer, K. S., Condon, E. P., and Howland, M. F.: Evaluation of wind resource uncertainty on energy production estimates for offshore wind farms, Journal of Renewable and Sustainable Energy, 16, 013 302, https://doi.org/10.1063/5.0166830, 2024.

Mortarini, L., Stefanello, M., Degrazia, G., Roberti, D., Trini Castelli, S., and Anfossi, D.: Characterization of wind meandering in low-wind-speed conditions, Boundary-Layer Meteorology, 161, 165–182, https://doi.org/10.1007/s10546-016-0165-6, 2016.

---

## Author Comment (AC2)

**Reply to the Editor's comment of "Brief communication: A note on the variance of wind speed and turbulence intensity"**

January 3, 2025

Please note that the Editor's comments are in *italic*, my responses in regular font, and the changes to the manuscript in blue color.

- *Line 15-20: The distinction between aligning the x-axis with the wind direction or with the East-West coordinate system is not unique to wind energy; it largely depends on the spatial and temporal scales of interest. In boundary-layer meteorology, particularly micrometeorology, the x-axis is typically aligned with the mean wind direction due to the focus on turbulence, as detailed in Kaimal and Finnigan (1994). In mesoscale meteorology, where the emphasis is on mean wind speed, the x-axis is, indeed, often aligned with the East-West direction. To avoid conflating discipline-specific conventions, I recommend acknowledging this broader context.*

  I agree that the convention of aligning the x-axis along the mean wind is not unique to the wind energy field. I added the following at line 13:

  This convention is also adopted in boundary-layer meteorology, particularly in micrometeorology, due to the focus on turbulence (Kaimal and Finnigan 1994).

  and the following at line 14:

  By contrast, in mesoscale meteorology and, more broadly, in geophysical applications, such as meteorological field campaigns or simulations of weather events, the convention is ...

- *I would go beyond the statement that the variance of the wind speed is often miscalculated. I would argue that using the variance of the wind speed itself—rather than treating the variance of the along-wind and cross-wind velocity components separately—is fundamentally problematic. In wind engineering and micrometeorology, these components are considered separately due to their distinct characteristics. The design of wind turbines, particularly for structural and turbulent loading considerations, is based on the variances of the along-wind and cross-wind components, not the wind speed. The continued use of wind speed variance might be a legacy of outdated practices.*

  *Line 28-29: The statement "turbulence intensity is a function of the standard deviation of wind speed" could be misleading. From micrometeorology and wind engineering perspectives, turbulence intensity is typically defined based on the individual velocity components (along-wind, cross-wind, and vertical), not wind speed. Defining turbulence intensity based on wind speed lacks physical relevance. In my humble opinion, its continued use in wind energy science is puzzling.*

  I agree with you on both statements, and that is partly why I wrote this note. Turbulence intensity to me does not make sense without specifying along which direction. And yet the IEC standard uses exactly the definition that you are referring to. As such I modified the text as follows:

  Since turbulence intensity is defined in the IEC standard as "ratio of the wind speed standard deviation to the mean wind speed" ...

  and

  It is important to note that the IEC standard is possibly the only case in which a single value of turbulence intensity is adopted. In most fields, three turbulence intensities are typically used, one for

each direction ($i_x = \sigma_u / \bar{U}$, and similarly for $i_y$ and $i_z$), where $x, y$, and $z$ are either the three Cartesian directions (e.g., in mesoscale meteorology) or the along-wind, cross-wind, and vertical directions (e.g., in micrometeorology, wind turbine design, and wind turbine load studies).

- *Line 31: While it is true that mesoscale meteorology often simplifies wind velocity as a 2D vector, this approach does not hold in micrometeorology or wind energy, where the vertical velocity component significantly contributes to turbulence kinetic energy (TKE). I understand that the inclusion of TKE in this discussion depends on the desired level of detail. If brevity is prioritized, this aspect could be omitted.*

Point well taken. I modified the notation in Sections 2 and 3 to be fully 3D. Then I introduced the simplification of a 2D vector at the end of Section 3, for the sake of simplicity and because the 2D approximation has often been used in wind energy applications. I believe the reason why the 2D approximation has often been adopted in wind energy is that cup anemometers have been historically used instead of sonics, and therefore it was not possible to measure the vertical component of the wind anyway. Here is the modified text:

To simplify the notation without loosing generality, we hereafter assume that the wind is a two-dimensional vector. This assumption is often used in mesoscale meteorology and is needed when only 2D measurements of the wind are available (e.g., with a cup anemometer). Thus all terms that are a function of $w$ drop from Eq. 17:

- *Conflict of Definitions in Different Fields: There may be conflicting definitions of "turbulence" between mesoscale and microscale meteorology that require clarification. In micrometeorology, turbulence is typically considered a three-dimensional process occurring within temporal scales of up to one hour and spatial scales smaller than a few kilometres. In micrometeorology, the variance of the along-wind and across-wind components differs significantly. Motions exceeding these scales are often classified as "non-turbulent motion," consistent with the concept of the spectral gap. However, mesoscale meteorology may occasionally describe such motions as "2D turbulence." These differences reflect divergent focuses and terminologies across disciplines and should be recognized explicitly.*

The reason why I introduced the simplification of a 2D wind vector in the first version of this note was that all the papers in the literature that have used the incorrect approximation were using it for 2D, i.e., they were summing the variances along $x$ and $y$ only. Since I made the change that you recommended and used the full 3D notation in the revised version, I do not think I should discuss the different definitions of turbulence across different fields here.

- *Table 1: The two examples in Table 1 effectively demonstrate the value of the proposed equation. However, it is unclear whether the statistics are based on six hours of data or shorter sub-samples. If turbulence is the focus, time averaging over six hours is not appropriate, especially since the second panel shows clear non-stationary fluctuations. If the table uses shorter intervals (e.g., 10 minutes to 30 min), I recommend expanding the analysis to include all samples from Figure 1. Comparing the wind speed variance estimated by the older equation with that from your proposed equation would strengthen the analysis. A scatter plot of these comparisons across the full dataset would complement Table 1. This visualization would make it easier to assess the overall performance and accuracy of the new equation relative to the older one.*

Thank you for pointing out that the second case was non-stationary and for explaining that a six-hour window for turbulence statistics is too long. I shortened the averaging window to three hours (instead of six) and found another case that was stationary (10/21/2016). All the statistics have been recalculated in Table 1.

As for the comment "Comparing the wind speed variance estimated by the older equation with that from your proposed equation would strengthen the analysis," I had already done this comparison in Table 1 (last 2 rows). To make it clearer, in the revised version I added the labels "Wrong $\sigma_U^2$" and "Wrong $TI$."

The data collected during the VERTEX campaign were post-processed and averaged over 5-minute intervals and those are the only data available. The original 20-Hz dataset was not retained unfortunately. As such, the scatter plot comparison that you are proposing is not possible.

---

## Author Comment (AC5)

**Reply to the Editor and Reviewers of "Brief communication: A note on the variance of wind speed and turbulence intensity"**

Cristina L. Archer

16 February 2025

Please note that the Reviewers' comments are in *italic*, my responses in regular font, and the changes to the manuscript in blue color.

**Reviewer # 1**

- *This paper deals with the difference between the variance of the wind component along the mean wind vector and the variance of the length of the wind vector, also called the wind speed. It is well known that those quantities are under most circumstances (i.e. not too high turbulence intensity) almost equal (e.g. L.. Kristensen 1998, JTech, vol 5, p6). The transverse component enters only the speed variance to second order in the turbulence intensity (see eq 8 in the mentioned paper). These observations do not change if the coordinate system is not aligned with the wind.*

The variance of the wind speed $\sigma_U^2$ and the variance of the component of the wind vector that is aligned along the mean wind direction $\sigma_u^2$ (if available), are indeed almost equal. For example, for the AWAKEN data that are discussed in the revised version of the manuscript, I calculated the variance of the x- and y-components after rotating the axes so that the x-axis would align with the mean wind direction, recalculated every 10 minutes. I found that $\sigma_u^2$ is indeed a very good approximation for the variance of the wind speed $\sigma_U^2$, with an average absolute percent error of 2.5%.

This fact is acknowledged in the manuscript:

"With this convention [to align the x-axis along the mean wind direction], the variance of wind speed is accurately approximated as the variance of the u-component of the wind, i.e., the component along x."

However, in the manuscript I am not talking about the "rotated" coordinate system, in which the x-coordinate is in alignment with the mean wind direction; I am talking about the geophysical coordinate system with coordinates aligned with the east-west (x), north-south (y), and vertical (z) directions, like in sonic anemometers (e.g., see the first sentence of the Definitions section: "Let us use the geophysical system of coordinates."). The rotated statistics are not always available and, to calculate them accurately, the raw data are needed. But, if the raw data are available, one might as well calculate the wind speed and its variance directly without bothering with the coordinate rotation.

The points I am trying to make are that:

1. an inaccurate equation has been used in the literature for cases with the **geophysical** coordinate system; and

2. the wrong equation has been used as the **definition** of the standard deviation of wind speed, whereas it just provides an approximation for it, and not a good one.

The paper by Kristensen (1998) deals with the calibration of cup anemometers in wind tunnels under steady-state conditions. Eq. 8 in particular is:

$$U = \sqrt{u^2 + v^2} \approx \bar{u} + u' + \frac{v'^2}{2\bar{u}}. \tag{1}$$

You cite this equation to support that transverse perturbations only affect the variance of wind speed to the second order; however, this is **not** an equation for the variance of wind speed, it is an equation for the wind speed, thus the point is not proven with this equation.

The following sentence was modified to clarify that the proposed equations are for the geophysical reference system:

"This note addresses this issue by proposing an analytical approximation for the wind speed variance and one for turbulence intensity for the geophysical system of coordinates."

- *The other subject paper is an apparent mistake in the literature. The author states that the variance of the wind speed is sometimes mistakingly said to be equal to the sum of the variances of the two horizontal components. This is obviously wrong, as the author clearly states, but I'm am unaware of these mistakes in the literature. The author does not provide evidence for these mistakes, which makes the need for this paper limited. The author might be wary to point out mistakes in specific papers, but this is unfortunately what has to be done in order to advance science. You cannot leave it to the readers to find documentation for this possible mistake in the literature.*

I am indeed uncomfortable publishing a note that directly points out mistakes by fellow scientists. In addition, it would take a huge effort to try to find all occurrences of the mistake in the literature. The point of my note is to provide a clear reference as to why the two variances are not the same. As such, I provide below a list of five papers with the above-mentioned error. My intention here is to satisfy your legitimate request for evidence, but I do not intend to add this list in the main document. Since the entire review process is public in WES, it will be possible in the future to find this information anyway, but, as far as I am concerned, not in the main manuscript.

- – Eq. 6 in Joffre and Laurila (1988);
- – Eq. 1 in Mortarini et al. (2016);
- – Eq. 1 in Lee and Lundquist (2017);
- – Eq. 1 in Bodini et al. (2020); and
- – Eq. 11 in Klemmer et al. (2024).

As you recommended in the online public discussion, I decided to add a citation to the oldest of the five papers above, i.e., Joffre and Laurila (1988) as follows:

"and often treated, incorrectly, as an exact definition (see for example Eq. 6 in Joffre and Laurila (1988))."

**Editor**

- *Line 15-20: The distinction between aligning the x-axis with the wind direction or with the East-West coordinate system is not unique to wind energy; it largely depends on the spatial and temporal scales of interest. In boundary-layer meteorology, particularly micrometeorology, the x-axis is typically aligned with the mean wind direction due to the focus on turbulence, as detailed in Kaimal and Finnigan (1994). In mesoscale meteorology, where the emphasis is on mean wind speed, the x-axis is, indeed, often aligned with the East-West direction. To avoid conflating discipline-specific conventions, I recommend acknowledging this broader context.*

I agree that the convention of aligning the x-axis along the mean wind is not unique to the wind energy field. I added the following at line 13:

This convention is also adopted in boundary-layer meteorology, particularly in micrometeorology, due to the focus on turbulence (Kaimal and Finnigan 1994).

and the following at line 14:

By contrast, in mesoscale meteorology and, more broadly, in geophysical applications, such as meteorological field campaigns or simulations of weather events, the convention is ...

- *I would go beyond the statement that the variance of the wind speed is often miscalculated. I would argue that using the variance of the wind speed itself—rather than treating the variance of the along-wind and cross-wind velocity components separately—is fundamentally problematic. In wind engineering and micrometeorology, these components are considered separately due to their distinct characteristics. The design of wind turbines, particularly for structural and turbulent loading considerations, is based on the variances of the along-wind and cross-wind components, not the wind speed. The continued use of wind speed variance might be a legacy of outdated practices.*

  *Line 28-29: The statement "turbulence intensity is a function of the standard deviation of wind speed" could be misleading. From micrometeorology and wind engineering perspectives, turbulence intensity is typically defined based on the individual velocity components (along-wind, cross-wind, and vertical), not wind speed. Defining turbulence intensity based on wind speed lacks physical relevance. In my humble opinion, its continued use in wind energy science is puzzling.*

  I agree with you on both statements, and that is partly why I wrote this note. Turbulence intensity to me does not make sense without specifying along which direction. And yet the IEC standard uses exactly the definition that you are referring to. As such I modified the text as follows:

  Since turbulence intensity is defined in the IEC standard as the "ratio of the wind speed standard deviation to the mean wind speed" ...

  and

  It is important to note that the IEC standard is possibly the only case in which a single value of turbulence intensity is adopted. In most fields, three turbulence intensities are typically used, one for each direction ($i_x = \sigma_u/\bar{U}$, and similarly for $i_y$ and $i_z$), where $x, y$, and $z$ are either the three Cartesian directions (e.g., in mesoscale meteorology) or the along-wind, cross-wind, and vertical directions (e.g., in micrometeorology, wind turbine design, and wind turbine load studies).

- *Line 31: While it is true that mesoscale meteorology often simplifies wind velocity as a 2D vector, this approach does not hold in micrometeorology or wind energy, where the vertical velocity component significantly contributes to turbulence kinetic energy (TKE). I understand that the inclusion of TKE in this discussion depends on the desired level of detail. If brevity is prioritized, this aspect could be omitted.*

  Point well taken. I modified the notation in Sections 2 and 3 to be fully 3D. Then I introduced the simplification of a 2D vector at the end of Section 3, for the sake of simplicity and because the 2D approximation has often been used in wind energy applications. I believe the reason why the 2D approximation has often been adopted in wind energy is that cup anemometers have been historically used instead of sonics, and therefore it was not possible to measure the vertical component of the wind anyway. Here is the modified text:

  To simplify the notation without loosing generality, we hereafter assume that the wind is a two-dimensional vector. This assumption is often used in mesoscale meteorology and is needed when only 2D measurements of the wind are available (e.g., with a cup anemometer). Thus all terms that are a function of $w$ drop from Eq. 17

- *Conflict of Definitions in Different Fields: There may be conflicting definitions of "turbulence" between mesoscale and microscale meteorology that require clarification. In micrometeorology, turbulence is typically considered a three-dimensional process occurring within temporal scales of up to one hour and spatial scales smaller than a few kilometres. In micrometeorology, the variance of the along-wind and across-wind components differs significantly. Motions exceeding these scales are often classified as "non-turbulent motion," consistent with the concept of the spectral gap. However, mesoscale meteorology may occasionally describe such motions as "2D turbulence." These differences reflect divergent focuses and terminologies across disciplines and should be recognized explicitly.*

I added a discussion on the time scales and disciplines, per your and Reviewer # 2' suggestion, as follows:

"The IEC definition of TI is also troubling because it does not specify which temporal scales should be considered in its calculation. Strictly speaking, turbulence intensity should refer only to fluctuations of the wind in the micro-scale (i.e., time averages of the order of minutes), thus to the right of the spectral gap in the wind spectrum. By contrast, wind fluctuations associated with meso or synoptic scale features belong to the left of the spectral gap and should not be called turbulent. In such cases, the ratio of the wind speed standard deviation over the mean, calculated over longer time intervals (i.e., hours to days), can still be obtained, but it should not be called a "turbulence" intensity. The equations derived here may be applied to any scale, but the focus is on the micro-scale."

- *Table 1: The two examples in Table 1 effectively demonstrate the value of the proposed equation. However, it is unclear whether the statistics are based on six hours of data or shorter sub-samples. If turbulence is the focus, time averaging over six hours is not appropriate, especially since the second panel shows clear non-stationary fluctuations. If the table uses shorter intervals (e.g., 10 minutes to 30 min), I recommend expanding the analysis to include all samples from Figure 1. Comparing the wind speed variance estimated by the older equation with that from your proposed equation would strengthen the analysis. A scatter plot of these comparisons across the full dataset would complement Table 1. This visualization would make it easier to assess the overall performance and accuracy of the new equation relative to the older one.*

Thank you for pointing out that the second case was non-stationary and for explaining that a six-hour window for turbulence statistics is too long. I was able to obtain another dataset, from the AWAKEN field campaign, which contains raw data at 20 Hz, and therefore I could calculate the 10-minute statistics directly and compare them against my formulas. I was able to prepare new figures with the scatter plots that you suggested. The message is even clearer now.

**Reviewer # 2**

- *In my opinion, this relates to 1) split disciplines between wind systems engineering and micrometeorology, and 2) a confusion between timescales (and intrisically linked space scales) in the literature and current practice. There I would first like to mention that while wind engineering is my background, I know only little about meteorology.*

I think that you "nailed" the issue perfectly. I was originally thinking that reason 1) was the main culprit, but your suggestion about time scales is very interesting.

- *Wind turbine structures are typically only concerned by microscale, and 10-minutes load cases are tradiationally used. There it is assumed that turbines would yaw to align with an assumed constant wind direction. Fluctuations are then represented around this direction and separated into along-wind and cross-wind drections. In this case the mean cross-wind component v_bar is always zero, and the original equation to compute TIs is valid. An exception may be for wake steering applications where a yaw misalignment with mean flow is intentionally created, but the coordinate system used to represent the wind speed is still relative to the slowly-varying wind direction.*

Even if the mean wind direction is used as the x-axis, and therefore $\bar{v}$ is zero, neither the original formula for TI (now in Eq. 9) nor that for $\sigma_U^2$ (now in Eq. 8) are valid. I think the origin of the error is in the calculation of $\sigma_U^2$, which is equal to the original wrong formula if and only if the x- and y-components are independent from each other and therefore the co-variances are zero; in other words, if turbulence is purely isotropic (never in the real atmosphere). Only in such a case would the original formula be correct.

Note that, with the x-axis aligned with the mean wind direction, an excellent approximation for $\sigma_U^2$ is actually $\sigma_u^2$ (not the original wrong formula in Eq. 8). The error is a few percent at most. I recognized this in the manuscript:

"With this convention [to align the x-axis along the mean wind direction], the variance of wind speed is accurately approximated as the variance of the u-component of the wind, i.e., the component along x."

- *However, longer load cases may be of interest when for instance looking at slowly-varying motions of floating substructures or power fluctuations from wind farms heavily influenced by mesocale fluctuations. In this case, the concept itself of characterising wind fluctuations by turbulence intensities covering all timescales (integrated over the entire width of the wind spectrum) is discussable, and might be outdated practice. Strictly speaking, TIs should only be used to describe microscale fluctuations (i.e what is commonly referred to as turbulence, to the right of the spectral gap in the wind spectrum), while mesoscale fluctuations (to the left of the spectral gap in the wind spectrum) should be described by a distinct quantity. Assuming "mesoscale turbulence intensities" are used for this purpose, I agree that they should be calculated using the method you suggest.*

The method that I suggest is valid for any scale, not just for the mesoscale. The one and only assumption is that the fluctuations should not be too large. However, I agree that the ratio of standard deviation of wind speed over mean wind speed should be called "turbulence intensity", as the IEC standard does, only if it refers to micro-scale turbulence.

In meso- or large-scale meteorology, there are obviously means and perturbations, but the latter are never referred to as turbulence, rather, as "eddy" or "transient" features. I have also seen "zonal" and "meridional" being used to indicate the mean flow (generally along the latitudinal zones) and the perturbation (generally north-south), respectively. The a-geostrophic wind, for example, is nothing but a perturbation of the wind around the geostrophic wind vector; yet, nobody would refer to it as turbulence.

I used a six-hour window to calculate the statistics in the figures included in the original manuscript. This was a poor choice on my side, which also the Editor pointed out, because it may have given the impression that my equations are only valid at the mesoscale and not at the micro-scale. I suspect you may have been confused by that too. In the revised version, I redid the analysis over 10-minute windows with another dataset, the AWAKEN field campaign, for which 20 Hz raw data were available.

- *To improve the quality and impact of your manuscript, I would suggest to 1) make this distinction between scales, disciplines and applications clearer, particularly through the role of turbine yawing*

I added a discussion on the time scales and disciplines, per your and the Editor's suggestion, as follows:

"The IEC definition of TI is also troubling because it does not specify which temporal scales should be considered in its calculation. Strictly speaking, turbulence intensity should refer only to fluctuations of the wind in the micro-scale (i.e., time averages of the order of minutes), thus to the right of the spectral gap in the wind spectrum. By contrast, wind fluctuations associated with meso or synoptic scale features belong to the left of the spectral gap and should not be called turbulent. In such cases, the ratio of the wind speed standard deviation over the mean, calculated over longer time intervals (i.e., hours to days), can still be obtained, but it should not be called a "turbulence" intensity. The equations derived here may be applied to any scale, but the focus is on the micro-scale."

- *2) add references to where you claim erroneous formulations have been used (coming from a different field, this statement looks superficial without examples)*

Also Reviewer #1 suggested that the papers where the issue was found should be listed, but I do not want to do it because, first, I do not want to "point the finger" at colleagues, and second, I cannot possibly provide a complete list. Here is a quick list of five papers:

  – Eq. 6 in Joffre and Laurila (1988);
  – Eq. 1 in Mortarini et al. (2016);
  – Eq. 1 in Lee and Lundquist (2017);
  – Eq. 1 in Bodini et al. (2020); and
  – Eq. 11 in Klemmer et al. (2024).

Note that it is WES policy that the review files remain publicly available, thus this list, if one really wanted, can always be found. As a compromise, Reviewer #1 suggested that I list only the very first paper in the main text, which I did as follows:

"and often treated, incorrectly, as an exact definition (see for example Eq. 6 in Joffre and Laurila (1988))."

**References**

Bodini, N., Lundquist, J. K., and Kirincich, A.: Offshore wind turbines will encounter very low atmospheric turbulence, Journal of Physics: Conference Series, 1452, 012 023, https://doi.org/10.1088/1742-6596/1452/1/012023, 2020.

Joffre, S. M. and Laurila, T.: Standard deviations of wind speed and direction from observations over a smooth surface, Journal of Applied Meteorology and Climatology, 27, 550 – 561, https://doi.org/10.1175/1520-0450(1988)027⟨0550:SDOWSA⟩2.0.CO;2, 1988.

Klemmer, K. S., Condon, E. P., and Howland, M. F.: Evaluation of wind resource uncertainty on energy production estimates for offshore wind farms, Journal of Renewable and Sustainable Energy, 16, 013 302, https://doi.org/10.1063/5.0166830, 2024.

Kristensen, L.: Cup anemometer behavior in turbulent environments, Journal of Atmospheric and Oceanic Technology, 15, 5–17, https://doi.org/10.1175/1520-0426(1998)015⟨0005:CABITE⟩2.0.CO;2, 1998.

Lee, J. C. Y. and Lundquist, J. K.: Evaluation of the wind farm parameterization in the Weather Research and Forecasting model (version 3.8.1) with meteorological and turbine power data, Geoscientific Model Development, 10, 4229–4244, https://doi.org/10.5194/gmd-10-4229-2017, 2017.

Mortarini, L., Stefanello, M., Degrazia, G., Roberti, D., Trini Castelli, S., and Anfossi, D.: Characterization of wind meandering in low-wind-speed conditions, Boundary-Layer Meteorology, 161, 165–182, https://doi.org/10.1007/s10546-016-0165-6, 2016.

---

## Author Response (AR2)

**Second reply to the Reviewers of "Brief communication: A note on the variance of wind speed and turbulence intensity"**

Cristina L. Archer

**9 April 2025**

Please note that the Reviewers' comments are in *italic*, my responses in regular font, and the changes to the manuscript in blue color.

**Reviewer # 1**

• Change "x-axis" to "\$x\$-axis, "u-component" to "\$u\$-component", etc.

Done.

• It would be nice with a column in table 1 for the special case of ave(v) = 0. Then the equations would mostly look like many people have seen them before.

Since Table 1 is calculated from the AWAKEN data, for which  $\bar{v} \neq 0$ , I should not do what this Reviewer is suggesting. In addition, the WES readers can easily figure out the final form of any equation in the manuscript for the case of  $\bar{v} = 0$ .

I think there is still a small issue with the equations. Eq (23) is slightly wrong. Consider the case where ave(v) = 0, v'= 0, ave(w) = 0 w'= 0 so we are left with ave(u) and the fluctuations in u, u'. Obviously, eqs (20), (22) and (23) are incorrect because they give ave(U) = ave(u)\*(1+1/2 sigma\_u^2 / ave(u)^2.

This brings me back to eq (1) in my first reviewer comment. If the average is taken of this equation, and you assume v'=0, then you end up with the correct result in this case, which is ave(U) = ave(u). Thus the approximation in (23) is not the best and that unfortunately propagates to the "(proposed)" results in table 1, only Eq 29 and Eq 23.

As it stands now, ave(U) approx  $ave(u)(1+1/2 (sigma_u^2 + sigma_v^2 + sigma_w^2)$ , which is not entirely correct. This should be fixed before publication of an otherwise valuable contribution.

First of all, Eq. 20 is absolutely correct in all cases with no exceptions because it is the result of simple mathematical steps without any assumption whatsoever.

Second, what the Reviewer is suggesting is a purely hypothetical case in which v' = 0 and w' = 0 at all times. This is a wind with a deterministic and fixed wind direction at all times, meaning that this wind vector has no fluctuations in direction (only in magnitude along x, the only relevant direction in this case). This situation is neither real nor realistic. In such an unrealistic case, then Eq. 22 does not reduce to  $\overline{U} = \overline{u}$ . However, this case cannot and will never happen in reality, thus I do not think it should be discussed in the manuscript.

However, I acknowledge that, if the axes are rotated in such a way that the x-axis is aligned with the mean wind direction, then the Kristernsen (1998)'s formula provides an excellent approximation, in particular also in the (unrealistic) case described above by the Reviewer. I added this at line 133:

"In this rotated coordinate system with the x-axis aligned with the mean wind, a better alternative to Eq. 23 for  $\overline{U}$  is the approximation from Kristensen (1998):

$$\bar{U} = \bar{u} + \frac{\sigma_v^2}{2\bar{u}^2}.$$
 (1)

**Reviewer #2**

• I think the essence of your answer "I think the origin of the error is in the calculation of  $\sigma_U^2$ , which is equal to the original wrong formula if and only if the x- and y-components are independent from each other and therefore the co-variances are zero; in other words, if turbulence is purely isotropic (never in the real atmosphere). Only in such a case would the original formula be correct." should stand in section 2 to help the reader understand. I would not use isotropic though, which does not mean uncorrelated.

I added the following at line 24:

"Note that  $\sigma_U^2$  would be equal to sum of the variances of the wind components if and only if the wind components were independent from each other and therefore their co-variances were zero. This, however, never happens in the real atmosphere."

• Else I agree with the editor's comment on the lack of physical sense of characterising turbulence using one TI based on U versus having different TIs for u, v and w as we would typically do in wind systems engineering. Now, that is in practice combined with an assumption of uncorrelated components which I agree is discussable. What would be needed is a covariance matrix.

I agree, but the IEC standard nonetheless applies.

---

## Author Response (AR3)

**Reply to the Editor's review of "Brief communication: A note on the variance of wind speed and turbulence intensity"**

**Cristina L. Archer**

**21 April 2025**

Please note that the Editor's comments are in *italic*, my responses in regular font, and the changes to the manuscript in blue color.

• About  $mean(v) \neq 0$

As the reviewer pointed out, the manuscript considers the case where  $mean(v) \neq 0$ . However, in boundary-layer meteorology, turbulence intensity is typically calculated after rotating the coordinate system into a wind-aligned frame, which by definition results in mean(v) = 0. From this perspective, performing such a rotation is a necessary step before calculating turbulence intensity, whether for an individual component or for the horizontal wind. To me, it would be unthinkable not to apply this rotation beforehand. But perhaps this approach is not as widely known outside the micrometeorology community as I assumed. Maybe this is another example of a clash between disciplines? In any case, this is indeed important to point this out.

I rewrote the introduction to better introduce the issue of the coordinate system and discuss how different disciplines use different axis conventions, which may have contributed to the confusion in the literature. Please see the paragraphs in the Introduction that start and end with:

"The first problem is the system of coordinates. [...] In summary, the relationship between the variance of wind speed and those of the wind components depends on the system of coordinates and therefore confusion can arise among disciplines because of their different axis conventions.

• About reviewer 2's question

Regarding your response to Reviewer 2 (on the correlation between the u and v components): It is true that u and v are not entirely uncorrelated. However, in most situations, the correlation is low enough that they are often treated as uncorrelated. I've only observed significant correlation in very specific environments, such as Norwegian fjords, where sonic anemometers were positioned near steep mountain slopes.

The wind components are weakly correlated in general, including in the AWAKEN dataset that I used in this manuscript. As you can see from Figure 1a and Table 1, when the covariance of u and v ( $\sigma_{uv}$ ) is accounted for (Eq. 27), the error is lower than when the covariance is ignored (Eq. 28). I agree that the error is small and in fact Eq. 28, which assumes that the wind components are uncorrelated, is a very good approximation.

However, what I stated in my reply ("Note that  $\sigma_U^2$  would be equal to the simple sum of the variances of the wind components ( $\sigma_u^2 + \sigma_v^2 + \sigma_w^2$ ) if and only if the wind components were independent from each other and therefore their covariances were zero.") is not true and therefore I removed it from the manuscript. It would be true if the wind speed magnitude was the linear sum of the two wind components; but it is a vector sum, thus a non-linear function (square root of the sum of the squares of the wind components). Using the sum of the wind component variances is just wrong in all situations, including when the two wind components might be entirely uncorrelated or even independent, because it has no theoretical or practical basis.

"There is no theoretical or statistical justification for this incorrect expression and no special case (e.g., independent or uncorrelated variables, or a specific statistical distribution, or particular spatial conditions) for which it would apply."

• Now to Reviewer 2's final comment:

"Else I agree with the editor's comment on the lack of physical sense of characterising turbulence using one TI based on U versus having different TIs for u, v, and w as we would typically do in wind systems engineering. Now, that is in practice combined with an assumption of uncorrelated components, which I agree is discussable. What would be needed is a covariance matrix."

I believe the answer regarding the applicability of the IEC standard is not entirely straightforward, as the IEC may be internally inconsistent. According to IEC Standard 61400-1, Equation (11), turbulence intensity is defined for the longitudinal component. In that sense, the standard does not contradict the reviewer's comment. More specifically,  $\sigma_1$  is defined as the standard deviation of the longitudinal wind component (Section 6.3), while  $\sigma_2$  and  $\sigma_3$  correspond to the lateral and vertical components, respectively. The subscript "1" clearly refers to the longitudinal component—not the horizontal component—as further supported by Equation (7) and the well-known "hypothesis of local isotropy in the inertial subrange." Importantly, this hypothesis does not apply to the horizontal wind component.

Nevertheless, as you correctly point out, the IEC also defines turbulence intensity more generally as "the ratio of the wind speed standard deviation to the mean wind speed, determined from the same set of measured data samples of wind speed, and taken over a specified period of time." To me, this reflects a significant inconsistency in the IEC's definition of turbulence intensity—one that may need to be addressed. If you agree, it might be worth highlighting this issue in your manuscript.

I reviewed the IEC standard carefully and I agree with you that it uses the rotated axis convention for which  $\sigma_{U}^{2} \approx \sigma_{1}^{2}$ . I expanded the discussion of the IEC standard in the Introduction as follows:

"The second problem is that of internal and external inconsistencies in the IEC standard. While the IEC standard clearly defines TI as the "the ratio of the wind speed standard deviation to the mean wind speed" in the "Terms and definitions" section, in later sections it actually appears to use  $\sigma_1$ , not  $\sigma_U$ , to define normal turbulence conditions and for fatigue load calculations (e.g., their Eq. 11). This would imply, wrongfully, that only the longitudinal fluctuations of the wind vector are relevant to a wind turbulence intensity is adopted. In most fields, including wind systems engineering, three turbulence intensities are typically used, one for each direction  $(TI_x = \sigma_u/\bar{U})$ , and similarly for  $TI_y$  and  $TI_z$ ). Lastly, the IEC standard assumes explicitly that the "turbulence standard deviation  $\sigma_1$  [...] shall be assumed to be invariant with height", while it is well known that there is a vertical gradient of TI in the atmospheric boundary layer, thus the turbulence fluctuations measured, for example, near the ground are not representative of those at hub height."